# Association of body index with fecal microbiome in children cohorts with ethnic–geographic factor interaction: accurately using a Bayesian zero-inflated negative binomial regression model

Jian Huang,[1,2] Yanzhuan Lu,[1,2] Fengwei Tian,[3,4] Yongqing Ni[1,2]

**ABSTRACT**   The exponential growth of high-throughput sequencing (HTS) data on the microbial communities presents researchers with an unparalleled opportunity to delve deeper into the association of microorganisms with host phenotype. However, this growth also poses a challenge, as microbial data are complex, sparse, discrete, and prone to zero inflation. Herein, by utilizing 10 distinct counting models for analyzing simulated data, we proposed an innovative Bayesian zero-inflated negative binomial (ZINB) regression model that is capable of identifying differentially abundant taxa associated with distinctive host phenotypes and quantifying the effects of covariates on these taxa. Our proposed model exhibits excellent accuracy compared with conventional Hurdle and INLA models, especially in scenarios characterized by inflation and overdispersion. Moreover, we confirm that dispersion parameters significantly affect the accuracy of model results, with defects gradually alleviating as the number of analyzed samples increases. Subsequently applying our model to amplicon data in real multi-ethnic children cohort, we found that only a subset of taxa were identified as having zero inflation in real data, suggesting that the prevailing understanding and processing of microbial count data in most previous microbiome studies were overly dogmatic. In practice, our pipeline of integrating bacterial differential abundance in microbiome data and relevant covariates is effective and feasible. Taken together, our method is expected to be extended to the microbiota studies of various multi-cohort populations.

**IMPORTANCE** The microbiome is closely associated with physical indicators of the body, such as height, weight, age and BMI, which can be used as measures of human health. Accurately identifying which taxa in the microbiome are closely related to indicators of physical development is valuable as microbial markers of regional child growth trajectory. Zero-inflated negative binomial (ZINB) model, a type of Bayesian generalized linear model, can be effectively modeled in complex biological systems. We present an innovative ZINB regression model that is capable of identifying differentially abundant taxa associated with distinctive host phenotypes and quantifying the effects of covariates on these taxa, and demonstrate that its accuracy is superior to traditional Hurdle and INLA models. Our pipeline of integrating bacterial differential abundance in microbiome data and relevant covariates is effective and feasible.

**KEYWORDS**   gut microbiome, children's cohort study, Bayesian regression, ZINB

H umans harbor an extremely complex gut microbiota whose composition diversifies between different populations. It plays a critical role in performing essential physiological functions for the life and health (1, 2). In recent decades, the development

**Peer Reviewers** Dong Chen, Shandong Sport University, Jinan, China; Luciana Damascena Silva, Instituto Evandro Chagas, Ananindeua, Pará, Brazil

Address correspondence to Yongqing Ni, niyqlzu@sina.com.

The authors declare no conflict of interest.

of high-throughput sequencing (HTS) technology has greatly advanced the progress of microbiology, shifting the perspective of microbiologists from culture to non-culture, and opening up new ways for us to understand trillions of microorganisms (3). Typically, microbiome data are obtained by amplifying and sequencing variable regions of the 16S rRNA gene from a few samples of interest, using HTS techniques. The results of dividing 16S rRNA genes into operational taxonomic units (OTU) or amplicon sequence variants (ASV) were summarized into multidimensional vectors of OTU/ASV counts for all samples. These count data provide information about the microbial composition and distribution of each sample with high resolution. Due to the continued explosion of HTS data on microbial communities, statisticians intend to develop a variety of methods to accurately characterize microbiome data, especially for complex microbial ecosystems, but the current statistical and analytical methods used remain challenging.

At first, statistical methods were used to explain differences in environmental factors by comparing the level of significance between different groups (environmental factors). However, since each covariate can only be analyzed separately as a categorical variable, these methods cannot handle the correlation between covariates well. Furthermore, evaluating the contribution of each covariates by regression analysis is a great strategy, which focuses on the comparison of microbiome community, such as the comparison of multi-taxa (4, 5), core microbiomes or enterotype, as well as the ratios (e.g., Firmicutes/Bacteroidetes ratio [6, 7]) of certain bacteria. However, these methods do not characterize differentially abundant species well, resulting in difficulties in clinical trials, mechanism validation, and biological replication. Another approach take each individual bacteria as a dependent variable for different subject groups or conditions, interrogates one-by-one. Unfortunately, one deficiency of these methods is that the optimization for microbiome count data sets has been ignored.

Current literature shows that due to the overdispersion, zero inflation, and fluctuating library size of the microbial count data (8, 9), the analysis of microbial data is complicated. Usually, in many microbiome data analyses, Gaussian distributions are used to simplify the calculation, without considering properties of the microbiome count data. In fact, using a Gaussian distribution model for regression analysis can lead to unreliable coefficient estimates (10), thereby undermining the reliability and accuracy of the model. The existing literature shows that negative binomial regression models are more suitable for a variety of data that exhibit a larger variance or/and overdispersion (11–14), whereas the Hurdle (15) and the zero inflation counting model have been found to be effective when processing count data having an excessive number of zero. Recently, based on the advantages and disadvantages of the above research models, zero-inflated negative binomial (ZINB) models were proposed by combining negative binomial model and zero inflation counting model. ZINB models are applicable when there is interest in a model for latent taxa corresponding to a impressionable microbiota at risk for the sequencing/PCR amplifying condition under study with counts generated from a negative binomial distribution and a non-impressionable microbiota that provides only zero counts. However, this methods are challenging to implement in real scenarios due to complex coding and theoretical obscurity. Indeed, it was sporadically applied to some specific research.

A growing number of studies have confirmed that some key taxa of the microbiome are closely associated with physical indicators (16–18) of the body, such as height, weight, age and BMI, which can be used as measures of human health. However, despite advancements in understanding the gut microbiota, it is unclear which aspects of ethnicity and geography (19, 20), whether culture-related activities (lifestyle) or genetics, underlie its observed association with the microbiome. In particular, accurately identifying which taxa in the microbiome are closely related to indicators of physical development is valuable as microbial markers of local child growth (21–23). In practice, due to the complexity of geographical factors and ethnic factors, as well as the difficulty of sampling, it is difficult to implement the research accurately.

In the current study, our goal was to verify the robustness and validity of our proposed zero-inflated negative binomial (ZINB) regression model by accurately identifying the normal developmental trajectory of the gut microbiota in growing children and the microbial taxa associated with body index, thereby optimizing a ZINB model aiming to figure out the complexity associated with zero inflation, overdispersion, and multivariate structure of gut microbiome data. To achieve this goal, the four children cohorts from two ethnic groups in Yili Prefecture, Xinjiang, western China, whose living habitat areas were non-overlapping and whose parental groups rarely intermarry across ethnic groups, were recruited. Clearly, geography represents an ensemble of environmental and cultural factors, including diet, lifestyle, and behavior habits. The idea is to reduce the complexity of the interplay between ethnic and geographic factors to facilitate the achievement of our goals. In the analysis process, our study has its advantages in the following treatment: (i) methods for analyzing microbiome data involving Bayesian variable selection strategies; (ii) constructing Bayesian regression based on the generalized linear model framework, which combines zero expansion, overdispersion, and multiple correlation structures; (iii) designing an analysis pipeline for microbial data with zero-inflation counting, including analysis of response variables one by one and taxonomic comparison, with special emphasis on the association between the microbiome data and covariates. This situation is consistent with the common idea of reducing the complexity of the interaction between ethnic and geographical factors in order to easily achieve our goals (24).

## Zero-inflated count data regression models

### Sampling model

The HTS data were analyzed to obtain a high-dimensional count matrix (OTU/ASV table). This represents every count of taxonomy bacterial species that the host individual contains. In this study, the count regression model is the preferred model of analysis. We assume that non-negative integer counts $Y_{ij}$ are observed for OTU/ASV $j$ in sample $i$, $j \in \{0, \ldots, J\}$ and $i \in \{1, \ldots, I\}$, and are organized in an $I \times J$ table, $Y = [Y_{ij}]$. In the zero-inflated negative binomial regression model, the crucial objectives are to make sure the significant factors could influence the $Y_{ij}$, and to determine the extent of the effect of potential host and environmental factors on the count of non-negative integer counts $Y_{ij}$. Specifically, we assume that each $y_j = Y_{ij}$ conforms to a separate zero-inflated count distribution as follows:

$$P(y_i) = \begin{cases} \phi + (1 - \phi)(\frac{\theta}{\theta + \lambda})^\theta, y_i = 0 \\ (1 - \phi)\frac{\Gamma(\theta + y_i)}{\Gamma(\theta)y_i!}(\frac{\theta}{\theta + \lambda})^\theta(\frac{\lambda}{\theta + \lambda})^{y_i}, y_i > 0 \end{cases} \tag{1}$$

where $\phi \in [0,1]$ is the weight of extra zeros generated from the sampling count missing, including biological zeros and technical zeros (25). Thus, the zeros of OTU/ASV counts include both fixed zeros from absence of species, and random zeros from unknown group membership of present subjects. Although the random zeros generated from present subjects have intrinsic interest, their mixture has been treat as a extremely convenient construct that came into fitting a statistical distribution of excess zeros in bacterial counts. $\lambda$ represents an average of $Y_{ij}$, and the dispersion parameter $\theta^{-1}$ is adjusted to a negative binomial distribution (NB) (so the variance is $\lambda + \frac{\lambda^2}{\theta}$) under the Poisson distribution (the variance is $\lambda$).

Thus, the joint distribution across all $i$ individuals in the sample based on equation 2 is:

$$f(y_i|\phi,\mu) = \prod_{y_i=0}\left[\left(\frac{\phi}{1-\phi} + \left(\frac{\theta}{\theta+\mu}\right)^\theta\right)(1-\phi)\right]$$
$$\prod_{y_i>0}\left[(1-\phi)\frac{\Gamma(\theta+y_i)}{\Gamma(\theta)y_i!}\left(\frac{\theta}{\theta+\mu}\right)^\theta\left(\frac{\mu}{\theta+\mu}\right)^{y_i}\right] \tag{2}$$

where $\phi = (\phi_1, \phi_2, \phi_3, ..., \phi_j)$, $\mu = (\mu_1, \mu_2, \mu_3, ..., \mu_j)$, and $\theta = (\theta_1, \theta_2, \theta_3, ..., \theta_j)$ are dispersion parametera that are assumed not to depend on covariates.

In the generalized linear models, $\log(\mu)$ and $\text{logit}(\phi)$ are transformations that successfully linearize Poisson means and Bernoulli probabilities through logit model (26) as follows equation (3):

$$\begin{aligned}
\log(\mu_j) &= B\beta \quad \& \quad \text{logit}(\phi_j) = -\tau B\beta \\
\text{or} \quad \log(\mu_j) &= B\beta \quad \& \quad \text{logit}(\phi_j) = G\gamma \\
\text{or} \quad \log(\mu_j) &= B\beta \quad \& \quad \text{logit}(\phi_j) = \theta
\end{aligned} \tag{3}$$

where $B$ is the designed covariate matrix, and $\beta$ is the regression coefficient vector. $\tau$ is defined as real-valued shape parameter (27), which implies that $\phi = (1 + \lambda^\tau)^{-1}$.

## Likelihood function establishment

In this situation, the overall mean of specific bacteria $\mu = E[y_i]$ is the primary interest. The vector parameter of $\log(\mu)$'s with the intercept $\beta_0$ included denoted by $\beta = (\beta_0, \beta_1, ..., \beta_{j-1})'$ represents the same overall effect of covariates on species counts increment as in ZINB regression. In other words, $\exp(\beta)$ represents the multiplicative increase in $\log(\mu)$ count for species in the overall counts corresponding to a one-unit increase in the covariate matrix $B$.

$\beta$ in a maximum likelihood framework as follows:

$$L_{\text{zinb}}(\beta, \tau, \theta|y) = \prod_{\text{all } y_i}\left(1 + e^{-\tau B\beta}\right)^{-1}$$
$$\prod_{y_i=0}\left\{e^{-\tau B\beta} + \left[1 + \frac{1}{\theta}\left(1 + e^{-\tau B\beta}\right)e^{B\beta}\right]^{-\theta}\right\}$$
$$\prod_{y_i>0} -\frac{\Gamma(y_i+\theta)}{y!\Gamma(\theta)}\left[1 + \frac{1}{\theta}\left(1 + e^{-\tau B\beta}\right)e^{B\beta}\right]^{-\theta} \tag{4}$$
$$\left[\frac{(1 + e^{-\tau B\beta}))e^{B\beta}}{\theta + (1 + e^{-\tau B\beta}))e^{B\beta}}\right]^{y_i}$$

The log-likelihood of the ZINB model is :

$$l(\beta, \tau, \theta|y) = -\sum_i \log(1 + e^{-\tau B\beta})$$
$$+ \sum_{y_i=0}\log\left\{e^{-\tau B\beta} + \left[1 + \frac{1}{\theta}(1 + e^{-\tau B\beta})e^{B\beta}\right]^{-\theta}\right\}$$
$$- \sum_{y_i>0}\log y_i! + \sum_{y_i>0}\sum_{k=0}^{y_i-1}\log(k+\theta)$$
$$- \sum_{y_i>0}\theta\log\left[1 + \frac{1}{\theta}(1 + e^{-\tau B\beta})e^{B\beta}\right] \tag{5}$$
$$+ \sum_{y_i>0}y_i\left[\log(1 + e^{-\tau B\beta}) + B\beta\right]$$
$$- \sum_{y_i>0}y_i\log\left[\theta + (1 + e^{-\tau B\beta})e^{B\beta}\right]$$

## Estimation of the distinct-parameters ZINB regression model

We need to specify a prior distribution for parameters in the model to obtain a Bayesian estimation of the unknown parameters in the ZINB model. A great result depends on establishing a good initial prior information as well as a formulation of an informative prior distribution (28). In this model, we assumed that the overdispersion parameter $\theta$ and shape parameter $\tau$ follow the gamma distribution $\theta \sim [\text{dgamma } (a = 0.001, b = 0.001)]$ and model coefficient parameter $\beta$ follows the normal distribution $(0, 10^{-6})$. Thus, the prior distribution formulation is as follows:

$$
\begin{aligned}
p(\beta, \tau, \theta) = &\prod_{y_i = 0} \frac{1}{b^a \Gamma(a)} \tau^{a-1} e^{-\frac{\tau}{b}} \\
&\prod_{\text{all} y_i} \frac{1}{\sigma_{\beta_i}\sqrt{2\pi}} e^{\frac{-(\beta_i - \mu_{\beta_i})^2}{2\sigma^2 \beta_i}} \\
&\prod_{\text{all} y_i} \frac{1}{b^a \Gamma(a)} \theta^{a-1} e^{-\frac{\theta}{b}}
\end{aligned}
\tag{6}
$$

The posterior distribution of parameters can be obtained by combining the likelihood function (equation 5) with the prior function (equation 6) as follows:

$$
p(\beta, \tau, \theta | B; y) = \frac{p(\beta, \tau, \theta | y; B) p(\beta, \tau, \theta)}{\int p(\beta, \tau, \theta | y; B) p(\beta, \tau, \theta) d(\beta, \tau, \theta)}
\tag{7}
$$

We used standard Markov chain Monte Carlo (MCMC) methods, which are available in JAGS using runjags package (29) in R software. The estimated effect size will be determined by the Bayesian estimated average. We will assess the uncertainty of the estimated effect size with credible intervals.

## MATERIALS AND METHODS

### Design and study population

This study was conducted in accordance with the guidelines of the Declaration of Helsinki (30) and was approved by the Ethics Committee of the First Affiliated Hospital of Shihezi University Medical School (2017-117-01). For school-aged children, we communicated with local authorities and school principals to get permission to sample. In addition, the researchers trained the teachers, and the teachers publicized the purpose and significance of our research. At the same time, informed consent forms were issued, which the students take home for parents to decide whether to sign or not. The following days, the participants had their height and weight measured and were inquired about their date of birth and recent dietary habits. After the survey, the participants were supplied with a stool sampler, ice bag, and aseptic bag, and were given comprehensive guidance on how to collect and preserve the samples. The criteria for inclusion in the study were as follows: (i) school-aged children, (ii) ethnic minorities, and both parents being ethnic minorities, (iii) able to provide informed consent, (iv) no antibiotics or other medications that could affect the composition of gut microbes that have been used in the past 6 months, (v) willing to provide regular stool samples as required by the study. The criteria for exclusion in the study were as follows: (i) individuals with severe digestive disorders (e.g., Crohn's disease, ulcerative colitis) or other chronic diseases (e.g., diabetes, heart disease), (ii) recent (within 6 months) use of antibiotics or other drugs that affect the gut microbiota, (iii) individuals who are unable to provide informed consent. We obtained a total of 656 samples from 8 June 2021, of which only 585 were used in this study. The survey was a population-based cross-sectional study that included four different geographic locations (four primary schools, two kindergartens), two distinct ethnic groups (Uyghur and Kazakh), and an age

range between 3.71 and 14.98 years. All the recruits were from rural locations. Despite the presence of numerous ethnic groups in Xinjiang, only the samples from the Uyghur and Kazakh populations met our study criteria, as the sample sizes from other ethnic groups ($n = 71$, five enthnic, median = 15) were too small to be included.

### Stool sample collection and transportation

In the preliminary investigative work, the sampling kits provided to specific participants were labeled uniquely and clearly. Also, the participants were supplied with a stool sampler, sterile gloves, and aseptic bag, and were given comprehensive guidance on how to collect and preserve the samples. The sampling process touches the environment as little as possible, and the sampler wears sterile gloves and takes samples with a sterile scraper. Approximately 5–10 g of fresh stool samples were collected from from the fecal middens of the participants by the participants themselves or their guardians and stored in a specially made sterile fecal sampling tube for effective storage under normal temperature conditions. After the collection, the sample tube was sealed with a sealing film, placed in a double-sterile plastic bag, and marked again. All the collected samples were temporarily stored in the portable vehicle refrigerator at −20℃ and would be transported back to the laboratory (Shihezi University Food Biotechnology Research Center) within 48 h and was stored at −80℃ until DNA extraction.

### DNA extraction and 16s rRNA sequencing

Prior to DNA extraction, the samples were thoroughly homogenized with sterile phosphate-buffered saline (PBS). Approximately 1 mL of the resulting mixture was subsequently aliquoted for DNA extraction. The stool homogenization procedures were conducted within a biosafety cabinet and completed within 30 min to ensure sample integrity and minimize contamination risks. Total genomic DNA was extracted from fecal samples using the E.Z.N.A. Soil DNA Kit (Omega Bio-tek, Norcross, GA, U.S.) according to manufacturer's instructions. Concentration and purity of extracted DNA were determined with TBS-380 and NanoDrop2000, respectively. DNA extract quality was checked on 1% agarose gel. DNA extract was fragmented to an average size of about 400 bp using Covaris M220 (Gene Company Limited, China) for paired-end library construction. Aired-end library was constructed using NEXTFLEX Rapid DNA-Seq (Bioo Scientific, Austin, TX, USA). Adapters containing the full complement of sequencing primer hybridization sites were ligated to the blunt-end of fragments. Paired-end sequencing was performed on Illumina Novaseq 6000 (Illumina Inc., San Diego, CA, USA) at Majorbio Bio-Pharm Technology Co., Ltd. (Shanghai, China) using NovaSeq Reagent Kits according to the manufacturer's instructions (https://www.illumina.com/). The extracted DNA was used as a template for PCR amplification of the V1–V3 region of bacterial 16S rRNA genes by HiSeq sequencing. In brief, samples amplified with "27f-YM" used forward primer YM-27F (5′-AGR GTT YGA TYM TGG CTC AG-3′) (31) and a reverse primer 534R (5′-ATT ACC GCG GCT GCT GG-3′). The PCR conditions were as follows: start at 95℃ for 4 min, 30 cycles denaturing at 95℃ for 30 s, annealing at 72℃ for 50 s, and final extension at 72℃ for 10 min.

### Bioinformatics and biostatistics

Paired-end sequencing reads were processed using a usearch (v11.0.667_i86linux64) pipeline (32). The entire workflow includes merging sequences, trimming barcodes and primers, conducting sequential quality control, and generating zero-radius operational taxonomic units (zOTUs), aka amplicon sequence variants (ASVs). Taxonomy of the ASVs was then predicted using the SINTAX algorithm (33), with the settings strand both and "sintax_cutoff" 0.8 using the SILVA SSU rRNA 138 database (34). Finally, an abundance table was generated using the "otutab" command by mapping the zOTUs obtained from filtered read.

We will use rarefaction to standardize the sequencing depth across samples. The rarefaction depth will be set to 5,000 sequences for each sample. For analyzing the

microbiota alpha and beta diversity, Shannon index, Bray–Curtis distance were estimated using vegan R package (35).

## Weight, height and BMI for age Z-scores

We separately calculated weight, height, and BMI (kg/m$^2$) z-scores (z-BMI) standardized on age and sex as indicated by WHO Growth Reference (2006) charts using the zscorer (v.0.3.1) (36) package in R. The z-BMI standard deviation is usually used for descriptive statistics to classify individuals as follows: normal (−2 sd and ≤2 sd), overweight (2 sd and ≤sd), and obese (>4 sd).

## STORMS checklist

This study has been completed according to the STORMS Checklist (DOI: https://doi.org/10.5281/zenodo.11127032).

## RESULTS

### Simulation data generation and description

In the simulation studies, it is important to have a clear understanding of the properties of both covariates and response variables before conducting the simulations. It is assumed that the microbial population follows a zero-inflated negative binomial distribution, and it has been established that there are six covariates that may have an impact on it. Some of these covariates may have interactions, and therefore we analyze them separately.

Response variable was as follows: count—bacterial counts for ASV table follows a negative binomial distribution.

Covariates were as follows:

- Age—a discrete variable with a range of (3.0~15.0)
- Sex—a classified variable with two sexes, male and female.
- Ethnic—classified variable with two enthicities, Uighur (U) or Kazakh (K)
- Geography—classified variable with four locations, S, W, H, and N.
- Height—a discrete variable with a range of (0.90~1.50 m)
- Weight—a discrete variable with a range of (17.0~27.0 kg)

"Eliminate effects of interaction" was as follows:

- BMI—a discrete variable equal to weight/(height)$^2$, to eliminate the interaction both weight and height.
- BMIAZ—BMI for age Z-scores, to eliminate the interaction of weight, height, and age.
- HAZ—height for age Z-scores,
- WHZ—weight for height Z-scores,
- WAZ—weight for age Z-scores,

The item "Eliminate effects of interaction" in the above list refers to eliminating the interactions by differentiating the Z-scores of the body index. The properties of covariates, such as their distribution and range of settings in which they are measured, are explained in detail.

In order to investigate the finite sample properties of the zero-inflated negative binomial (ZINB) model, simulations were conducted focusing on count data regression. Let $Y_i$ represent the bacterial counts for the $i$th sample, where $x_{i1}$ corresponds to age ranging from 3.0 to 15.0 years, $x_{i2}$ denotes ethnicity (binary), $x_{i3}$ indicates geographic location (four categories), and $x_{i4}$ represents the body mass index for age Z-score (BMIAZ) with a mean of 0 and a standard deviation of 1. The data were generated using the R programming environment, with all variables adhering to the distributional

assumptions of the ZINB model. The simulation study considered sample sizes of 100, 200, 500, and 1,000. Correlation coefficients were categorized into three levels: low (0.5), moderate (1), and high (1.5). Zero inflation was also classified into three levels: no signal (0), moderate (0.2), and high (0.5). Over-dispersion ($1/\theta$), was examined at three levels: no signal (0), moderate (1), and high (2).

Each simulation scenario consisted of the following steps:

i. Take random samples from each data set with sizes of 100, 200, 500 and 1,000;
ii. Set iteration parameters as adapt = 10,000, burning = 2,000, sample = 2,000;
iii. Fit 10 models:
 a. Poisson (P) model
 b. Negative binomial (NB) model
 c. ZIP($\tau$) model
 d. MZIP model
 e. ZIP(inter) model
 f. ZINB($\tau$) model
 g. MZINB model
 h. ZINB(inter) model
 i. Hurdle model (37)
 j. INLA model (38)
iv. Record all simulation parameters and estimate their deviations.

### Z-score processing is better to eliminate interaction between covariables

In the experiment, the interactions brought about by covariates are often uncertain. We try to eliminate the interaction between covariates by using several processing method as Table 1 summarizes. We interact with generalized linear models (GLMs) for covariables that we suspect may interact, such as ethnicity and geography (maxGVIF = 125.87); height and weight (maxGVIF = 125.76). The experiment results show that GLM does not eliminate interactions very well. Next, we tried to replace weight and height with BMI, and GVIF value decreased significantly (maxGVIF = 16.40). The results show that interaction is reduced significantly, but it still does not reach the conditions for naive bayesian (GVIF < 3) (39, 40). This can indicate that the BMI index tends to change with age. Finally, we use BMI for age Z-score index to replace height * weight interaction. Interaction is eliminated successfully (maxGVIF = 2.12).

### The simulation experiments

We performed a series of simulations to systematically evaluate the performance of ZINB models. To verify the robustness of ZINB models to potential confounders, we compared their accuracy with that of seven other models, each subjected to varying levels of potential confounding factors (sample size, zero-inflated probability, overdispersion, and covariate correlation coefficient). Poisson (P) model is a Poisson regression model using Bayesian method that does not consider zero

**TABLE 1** Fits of some ZINB models for the number of bacterial counts

| Highest interactions in the model | Log-likelihood | Residual degrees of freedom | GVIF$(1/(2*Df))^{a}$ |
|---|---|---|---|
| Non-interactions | −6,985 | 991 | (1.00~125.73) |
| Ethnic * geography | −6,984 | 988 | (1.00~125.87) |
| Height * weight | −6,985 | 990 | (1.00~125.76) |
| Ethnic * geography + height * Weight | −6,983 | 987 | (1.00~125.89) |
| Ethnic * geography + WHZ | −6,985 | 989 | (1.00~9.12) |
| Ethnic * geography + height * WAZ | −6,984 | 988 | (1.00~106.28) |
| Ethnic * geography + age * BMI | −6,985 | 989 | (1.00~16.40) |
| Ethnic * geography + age * height * weight | −6,982 | 984 | (1.00~2114.55) |
| Ethnic * geography + BMIAZ | −6,985 | 989 | (1.00~2.12) |

$^{a}$Generalized variance inflation factor (GVIF) (41) is used to measure the influence of collinearity on the square of joint confidence region length (two or more coefficients).

inflation and overdispersion of counting data. The negative binomial (NB) model is a negative binomial regession model using Bayesian method that only considers overdispersion. *_m, *_inter and *_$\tau$ models respectively followed the link function $\text{logit}(\phi) = -\gamma_0 + \gamma_1 * x_1 + \gamma_2 * x_2 + \gamma_3 * x_3$, $\text{logit}(\phi) = \theta$ guides with $\theta$ belong to Gaussian distribution, $\text{logit}(\phi) = -\tau(\beta_0 + \beta_1 * x_1 + \beta_2 * x_2 + \beta_3 * x_3)$. The Hurdle model uses maximum likelihood estimation (MLE) for ZINB regression, and follows the link function $\text{logit}(\phi) = -\gamma_0 + \gamma_1 * x_1 + \gamma_2 * x_2 + \gamma_3 * x_3$. The INLA model is a Bayesian model performing ZINB regression, and follows the link function $\text{logit}(\phi) = \theta$ with $\theta$ obeying a gamma distribution.

We performed a series of simulations to systematically evaluate the performance of ZINB models. To verify the robustness of ZINB models to potential confounders, we compared their accuracy with that of seven other models, each subjected to varying levels of potential confounding factors (sample size, zero-inflated probability, overdispersion, and covariate correlation coefficient). Poisson (P) model is a Poisson regression model using bayesian method that does not consider zero inflation and overdispersion of counting data. Negative binomial (NB) model is a negative binomial regression model using Bayesian method that only consider overdispersion. *_m, *_inter and *_$\tau$ models respectively followed the link function $\text{logit}(\phi) = -\gamma_0 + \gamma_1 * x_1 + \gamma_2 * x_2 + \gamma_3 * x_3$, $\text{logit}(\phi) = \theta$ guides with $\theta$ that belong to Gaussian distribution, $\text{logit}(\phi) = -\tau(\beta_0 + \beta_1 * x_1 + \beta_2 * x_2 + \beta_3 * x_3)$. The Hurdle model uses maximum likelihood estimation (MLE) for ZINB regression and follows the link function $\text{logit}(\phi) = -\gamma_0 + \gamma_1 * x_1 + \gamma_2 * x_2 + \gamma_3 * x_3$. The INLA model is a Bayesian model performing ZINB regression and follows the link function $\text{logit}(\phi) = \theta$ with $\theta$ obeying a gamma distribution.

Initially, we conducted a comparative analysis of the performance metrics across ten distinct models through estimated accuracy by reporting the absolute relative bias (ARB). Figure 1a illustrates the median ARB of the estimated simulated effect sizes across all scenarios. The four zero-inflated negative binomial (ZINB) models demonstrated superior accuracy relative to the Poisson model, with the ZINB_tau model exhibiting the highest precision, as indicated by a median ARB of 0.55. However, the Hurdle model is insufficient in regression analysis when dealing with overdispersion and an excessive number of zeros. Furthermore, we conduct a detailed evaluation of the accuracy of various models across differing levels of dispersion and zero inflation as shown in Fig. 1b. In simulations without dispersion and zero inflation, all models showed similar performance. All model accuracy decreased with the increase of extra zeros. ZINB model shows good performance when the degree of dispersion is increasing. Additionally, as depicted in Fig. 1b, we perform a comprehensive evaluation of the accuracy of various models across different levels of dispersion and zero inflation. In simulations devoid of dispersion and zero inflation, all models exhibited comparable performance. However, model accuracy generally declined with an increase in the proportion of excess zeros. Notably, the ZINB models demonstrated robust performance as the degree of dispersion increased. The results are similar for different covariate correlation coefficients (see Fig. S1 to S3; Table S1 to S3).

### Test ZINB models of tolerance for overdispersion

Although Agresti et al. (42) elucidated the concept of discretization in terms of the conditional variance and mean of count data, there remains obscurity regarding the impact of dispersion degree on the accuracy of count models. Several simulation studies (12, 43, 44) have explored the accuracy of the zero-inflated negative binomial models under conditions of excessive dispersion. However, these studies tend to underestimate dispersion values ($1/\theta < 2$), which is inconsistent with empirical observations, indicating that only rare taxa exhibit dispersion degrees below this threshold. To assess the impact of overdispersion on model performance, we adhere to the parameter settings outlined in the preceding subsection and evaluate model robustness by varying the

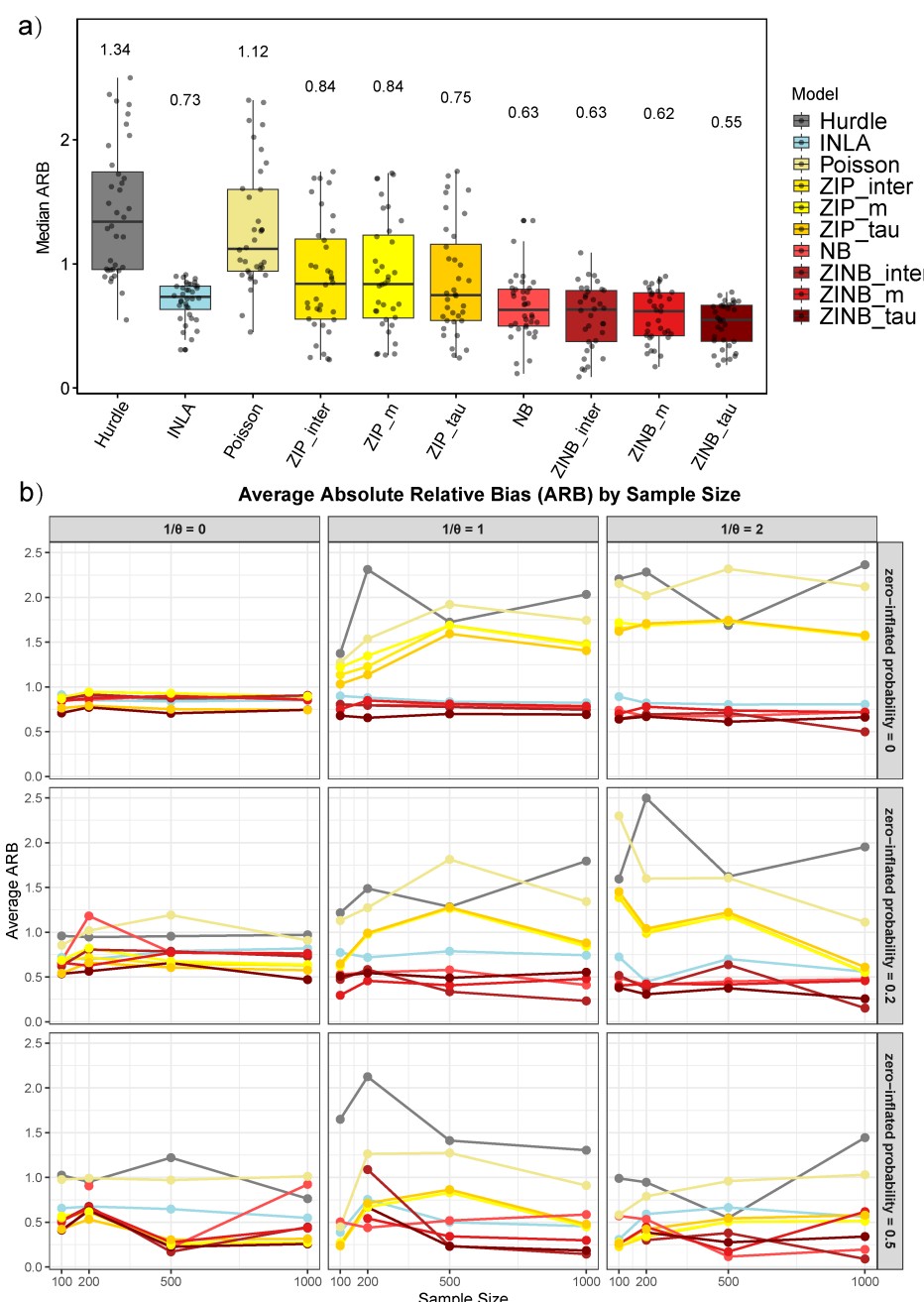

**FIG 1** Fits of 10 models in simulation experiment of bacterial counts. (a) The box plot illustrates the fitted ARB values for the 10 models across all scenarios, with distinct colors representing different models—red for ZINB models and yellow for ZIP models. The value displayed above each box plot denotes the median ARB for each model. (b) The fitting accuracy of 10 models in different cases, where each row corresponds to a distinct level of dispersion (null $1/\theta = 0$, median $1/\theta = 1$ and high $1/\theta = 2$), and each column represents a different probability of zero inflation (null 0, median 0.2, and high 0.5).

overdispersion parameters. We set five overdispersion parameters ($1/\theta = 2, 5, 10$) and add the sample size set (2,000) to simulate the model's response to excessive dispersion. Table 2 shows the ARB (SD) of the estimated simulated effect size for ZINB_inter, ZINB_m, ZINB_$\tau$, INLA, and Hurdle. For all scenarios, the ZINB_$\tau$ model provides more accurate estimates than other models in cases of overdispersion, as indicated by its lower ARB values. Moreover, we have observed that the presence of overdispersion prevents the model's accuracy from improving, even with an increase in sample size. Furthermore, in

cases of extreme overdispersion (with $1/\theta = 5$), partial models exhibit unestimable parameters (with the Slicer remaining fixed at a value with infinite density) for sample sizes below 500; conversely, in instances of overdispersion (with $1/\theta =10$), all models demonstrated unestimable parameters for sample sizes below 500 (Table 2). Aiming at these problems, Zitzmann et al. (45) reported and raised a question as to whether bias is the most important criterion to consider when the sample size is small. Therefore, it is obvious that people agree that large sample is the most effective method.

## Real data application

In our study, information of gut microbiome and body development index were obtained from a total of 585 healthy preschool and school-age children from Ili district of the westernmost part of Xinjiang, China. They were recruited from four different villages (four primary schools and three kindergartens) and mainly belong to two ethnic groups, Uyghur and Kazakh, as shown in Fig. 2a. Overall, the cohort of children in each village was overwhelmingly dominated by one of the ethnic groups; every cohort has an even ratio of girls to boys (see at https://doi.org/10.5281/zenodo.11127871). The gut microbiota of the children was examined through 16s amplicon sequencing. After removing noise from the 585 samples, we obtained 10,065 ASVs (see at https://doi.org/10.5281/zenodo.11127871). Firstly, we figured up measures of gut bacterial alpha diversity, including Shannon's index. We did not observe marked ethnic differences in Shannon's index ($P > 0.05$, Wilcoxon test), only the measurements from H and S villages show slight differences in Shannon's index ($P = 0.029$) as shown in Fig. 2b. In Fig. 2c, obesity-related differences are shown ($P > 0.05$, Wilcoxon test). Then, we performed statistics on bmiAgeZ for children cohorts of different ethnics in four villages, and the resulting date indicated the H village children bmiAgeZ was significantly higher than other groups ($P = 1.1 \times 10^{-12}$, Wilcoxon test), as shown in Fig. 2d. Shannon also showed no difference ($P > 0.05$, Tukey's test) in age gradient (Fig. 2e). However, principal coordinates analysis (PCoA) clearly demonstrated different population community distinctions between BMI or age groups (Fig. 2f and g). According to the results of the PERMANOVA analysis, age and BMI index were found to significantly impact the demarcation of the microbial community among children cohorts (age: $R^2 = 0.013$, $P < 0.001$; BMI: $R^2 = 0.006$, $P < 0.001$).

**TABLE 2** The average and standard deviation among five models in the simulation study[a,b]

| Overdispersion | n | Hurdle | | INLA | | ZINB_inter | | ZINB_m | | ZINB_τ | |
|---|---|---|---|---|---|---|---|---|---|---|---|
| | | 1//θ | ARB | 1//θ | ARB | 1//θ | ARB | 1//θ | ARB | 1//θ | ARB |
| 2 | 100 | 0.78 (0.78) | 5.07 (24.19) | 0.7 (0.31) | 0.77 (0.27) | 3.07 (5.1) | 0.72 (0.14) | 3.51 (13.65) | 0.72 (0.13) | 0.7 (1.6) | 0.7 (0.13) |
| | 200 | 1.45 (1.45) | 3.81 (8.62) | 1.8 (0.49) | 0.8 (0.37) | 4.09 (22.61) | 0.8 (0.1) | 4.08 (22.02) | 0.8 (0.1) | 2.06(8.59) | 0.77 (0.1) |
| | 500 | 0.28 (0.28) | 1.3 (3.15) | 1.45 (0.39) | 0.54 (0.25) | 2.31 (9.24) | 0.87 (0.06) | 2.72 (25) | 0.87 (0.06) | 1.71 (7.23) | 0.83 (0.06) |
| | 1,000 | 3.01 (3.01) | 1.75 (9.17) | 2.67 (0.4) | 0.79 (0.38) | 4.26(5.5) | 0.72 (0.04) | 4.27 (56.28) | 0.72 (0.03) | 1.65 (12.55) | 0.7 (0.03) |
| | 2,000 | 2.02 (2.02) | 5.32 (22.24) | 1.95 (0.23) | 0.79 (0.39) | 2.23 (5.05) | 0.74 (0.03) | 2.58 (8.81) | 0.72 (0.03) | 1.41 (18.14) | 0.82 (0.03) |
| 5 | 100 | NA | NA | 0.02 (0) | 0.41 (0.32) | 4.16(8.74) | 0.87 (0.13) | NA | NA | 0.74 (2.07) | 0.83 (0.13) |
| | 200 | NA | NA | 0.04 (0.02) | 0.4 (0.24) | 7.81 (7.7) | 0.27 (0.05) | 7.43 (16.9) | 0.27 (0.05) | 7.6(15.26) | 0.27 (0.05) |
| | 500 | 1.65 (1.65) | 6.21 (18.62) | 2.26(0.6) | 0.52 (0.2) | 7.07 (8.09) | 0.31 (0.03) | 7.92 (13.78) | 0.31 (0.03) | 6.67 (8.09) | 0.31 (0.03) |
| | 1,000 | 1.95 (1.95) | 2.9 (8.86) | 3.17 (0.7) | 0.58 (0.22) | 2.56(6.68) | 0.13 (0.02) | 8.5 (16.69) | 0.44 (0.03) | 7.3 (13.92) | 0.14 (0.02) |
| | 2,000 | 3.24 (3.24) | 3.78 (17.61) | 2.66(0.45) | 0.59 (0.25) | 6.5 (54.78) | 0.69 (0.03) | 8.44 (12.64) | 0.53 (0.02) | 6.61 (16.95) | 0.13 (0.01) |
| 10 | 100 | NA | NA | NA | NA | NA | NA | NA | NA | NA | NA |
| | 200 | NA | NA | NA | NA | NA | NA | NA | NA | NA | NA |
| | 500 | 0.18 (0.18) | 0.82 (4.62) | 0.03 (0.02) | 0.2 (0.11) | 2.82 (6.1) | 0.37 (0.04) | 12.82 (20.97) | 0.47 (0.06) | 15.65 (54.35) | 0.1 (0.01) |
| | 1,000 | 5.82 (55.82) | 5.88 (33.07) | 2.63 (0.62) | 0.5 (0.19) | 15.8 (34.01) | 0.34 (0.03) | 14.15 (17.47) | 0.24 (0.03) | 12.52 (13.25) | 0.19 (0.02) |
| | 2,000 | 5.49 (5.49) | 4.18 (26.44) | 3.55 (0.7) | 0.5 (0.19) | 6.45 (15.77) | 0.24 (0.02) | 11.89 (15.65) | 0.25 (0.02) | 9.24 (11.62) | 0.24 (0.02) |

[a]The 1/θin the first column is the parameter set by the model, and the 1/θin the following columns is the simulation result. ARB (sd) represents the average difference between the actual value (standard deviation) of each fitting result and the real value.
[b]"NA" indicates that the mode can not calculate the parameter result (Slicer stuck at value with infinite density).

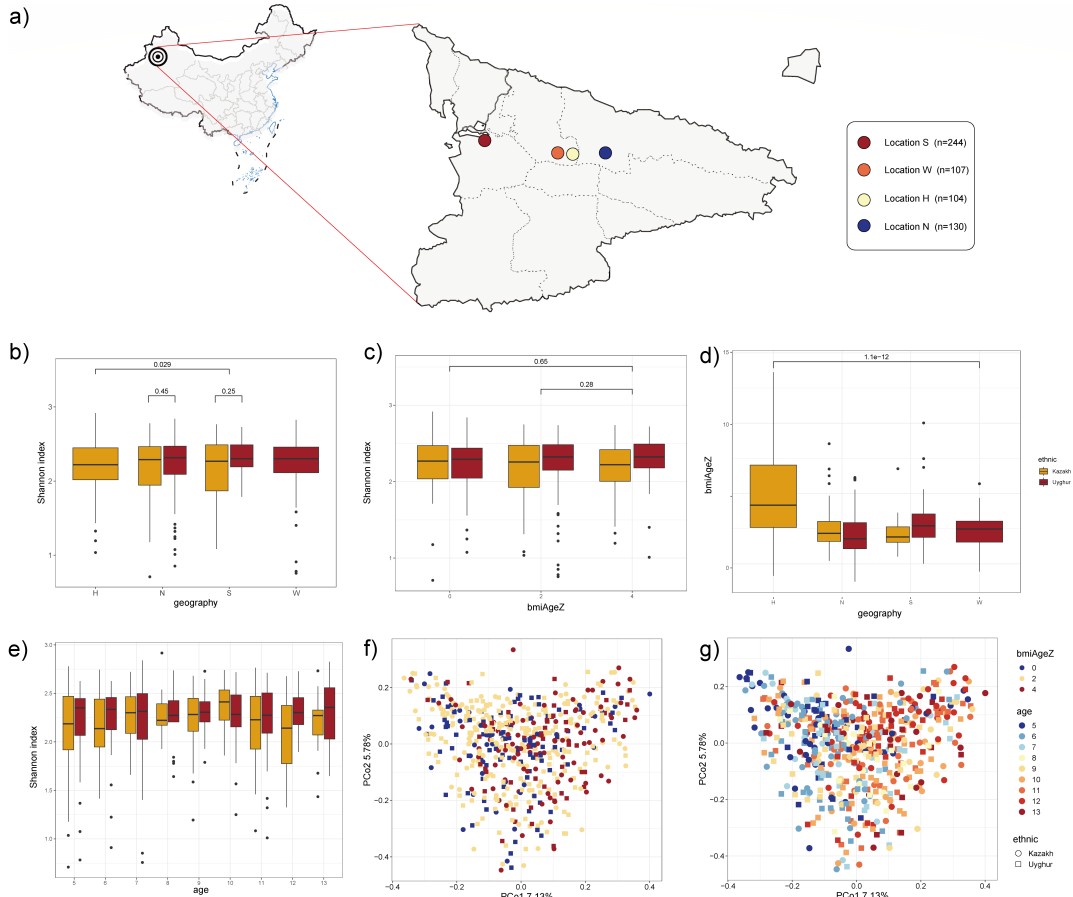

**FIG 2** Multi-ethnic cohort of children community $\alpha$-diversity and $\beta$-diversity. (a) Four locations of sampling Ili Kazak Autonomous Prefecture in Xinjiang, China; (b, c, e) $\alpha$-Diversity (Shannon diversity) of microbial communities in geography, BMI, and age based on V1 and V3 regions of 16S rRNA genes, respectively. The statistical method used is the Wilcoxon test performed in R, and the *P*-value is directly labeled above the box; (d) BMI differences in the ethnic and geographical distribution; (f,g) $\beta$-diversity across BMI and age, principal coordinates analysis (PCoA) ordination of bacterial communities based on the Bray–Curtis dissimilarity.

## *Interpretation of bacterial count data*

In this subsection, we discuss in detail about bacterial count properties. Initially, we excluded ASVs with counts less than 15 reads due to interference for the accuracy of the calculation. Since the high overdispersion parameters and zero inflation probabilities can significantly impact the accuracy of the model, we conducted an evaluation of the count data characteristics within this study cohort. Considering a hypothesize that the zero-inflated probability ($\phi$) and overdispersion parameters ($\frac{1}{\theta}$) follow a Gaussian distribution, we use it to calculate this cohort means. But due to the overdispersion parameters and the zero-inflation probability cannot be easily measured, we instead use an approximation estimation algorithm (see Appendix S1) to measure them. We figure up the original discrete parameters of each ASV and the overdispersion parameters of the part of the negative binomial distribution. By calculating the raw data directly, we observed that only 32% of the ASV data had zero-inflated. In ASVs with zero inflation, most of the zero expansion probabilities ($\phi$) are very high (mean = 90.2%). Furthermore, by calculating the overdispersion parameter of the negative binomial part (excluding the part of "extra-zero"), we find that the overdispersion parameter of the negative binomial part is slightly lower than that of the whole part (Fig. 3a), thus showing that zero inflation has a slight influence on the degree of data dispersion. Then, we measure the distribution of each ASVs by the mean value and standard deviations. All ASVs deviate from Poisson distribution (red solid line), instead exhibit dispersion phenomena,

and approximately 22.56% of ASVs showed overdispersion (with $\frac{1}{\theta}$ > 10) (illustrating in Fig. 3b). This indicates that microbial count data significantly deviate from the Poisson distribution, making it inadvisable to choose the Poisson model for regression analysis of such data. To enhance the application of the ZINB model, we examine the influence of taxon size on two critical parameters: overdispersion size and zero-inflated probability. These parameters serve as quantifiable measures for assessing overdispersion and zero inflation, respectively. Figure 3c and d visually presents the overdispersion and zero-inflation parameters across various taxonomic levels. Notably, while the probability of zero inflation progressively decreases with an increase in taxonomic rank, the overdispersion parameter is observed to be lowest at the ASV level. Given the greater influence of discrete parameters on the model relative to the zero-inflated probability, we have selected the ASV classification level for subsequent calculations.

### Application to multi-ethnic children cohort study

The Bayesian ZINB model is capable of conducting integrative analysis on microbiome count data. In this subsection, we applied this model to analyze microbiome data from a multi-ethnic children cohort study. The full data set includes 585 samples. Here, we aim to identify unique taxonomic signatures of the microbiome in children based on their age and BMI. At the species taxonomic level, the observed microbial abundance matrix Y was profiled from the gut microbiome. Before the downstream analysis, we filtered out ASVs with extremely low abundance (less than 30 observed counts in the cohort), following the suggestion in Wadsworth (46). We obtained 2,418 taxa for further analysis. To aid convergence for covariate information, the age variable should be centered, and the BMI Z-score variable should be used. Using a Bayesian ZINB regression model, we systematically examined each of the 2,418 taxa (ASVs) individually, considering the effects of ethnicity and geography while accounting for the interaction between these two factors.

Our model identified seriatim the age-discriminatory or BMI-discriminatory differentially abundant taxa (ASVs), and revealed the correlation between microbial groups and host physical indexes. We conducted a comparative analysis of the fitting results from five models applied to a multi-ethnic cohort of children. The findings indicated that all ZINB models exhibited consistency, with the ZINB model showing a minor discrepancy relative to the INLA model, while demonstrating a significant divergence from the Hurdle model (Fig. S4 and S5). Figure 4a shows the posterior mean of $\mu Y_{ij}$ for all discriminating taxa identified. We have shown that 160 ASV abundance remains age-discriminatory (as indicated by red dots), and 162 ASV abundance remains BMI-discriminatory (as indicated by blue dots). For children, the average abundance of *Bacteroides*, *Roseburia*, *Faecalibacterium*, *Streptococcus*, and *Actinomyces* decreases as age increases (47–49). On the other hand, *Catenibacterium*, *Bifidobacterium*, and *Ligilactobacillus* show an increase in average abundance with age (50). Additionally, the average abundance of Blautia decreases with an increase in BMI for children (51), while *Actinomyces*, *Roseburia*, *Blautia*, *Faecalibacterium*, *Coprococcus*, *Bacteroides*, *Streptococcus*, and *Ligilactobacillus* show an increase in average abundance with an increase in BMI for children (50, 52–54). Notably, *Ligilactobacillus* (ASV4) exhibited significant impact and low variability for both age and BMI, and possesses lower levels of dispersion ($1/\theta = 2.09$) and higher relative abundance (0.0372) (Fig. 4b). Some taxa are linked to both age and BMI, yet display completely opposite effect sizes, such as *Faecalibacterium* (ASV2580, ASV3278), *Streptococcus* (ASV3022), *Roseburia* (ASV3756), *Actinomyces* (ASV 4128), and *Bacteroides* (ASV8238) (Table S4 at https://doi.org/10.5281/zenodo.11127871). In our microbita count data, only a limited number of ASVs demonstrate zero inflation and low dispersion (Fig. 4b). Several studies have indicated a significant correlation between elevated levels of *Roseburia*, *Streptococcus*, and *Prevotella* with obesity. However, in other studies, conflicting outcomes were observed regarding the prevalence of *Anaerotruncus*, *Bacteroides*, *Blautia*, *Clostridium*, *Coproccocus*, *Faecalibacterium*, and *Ruminoccocus* in obese individuals. Indeed, the current studies show no clear conclusions about the

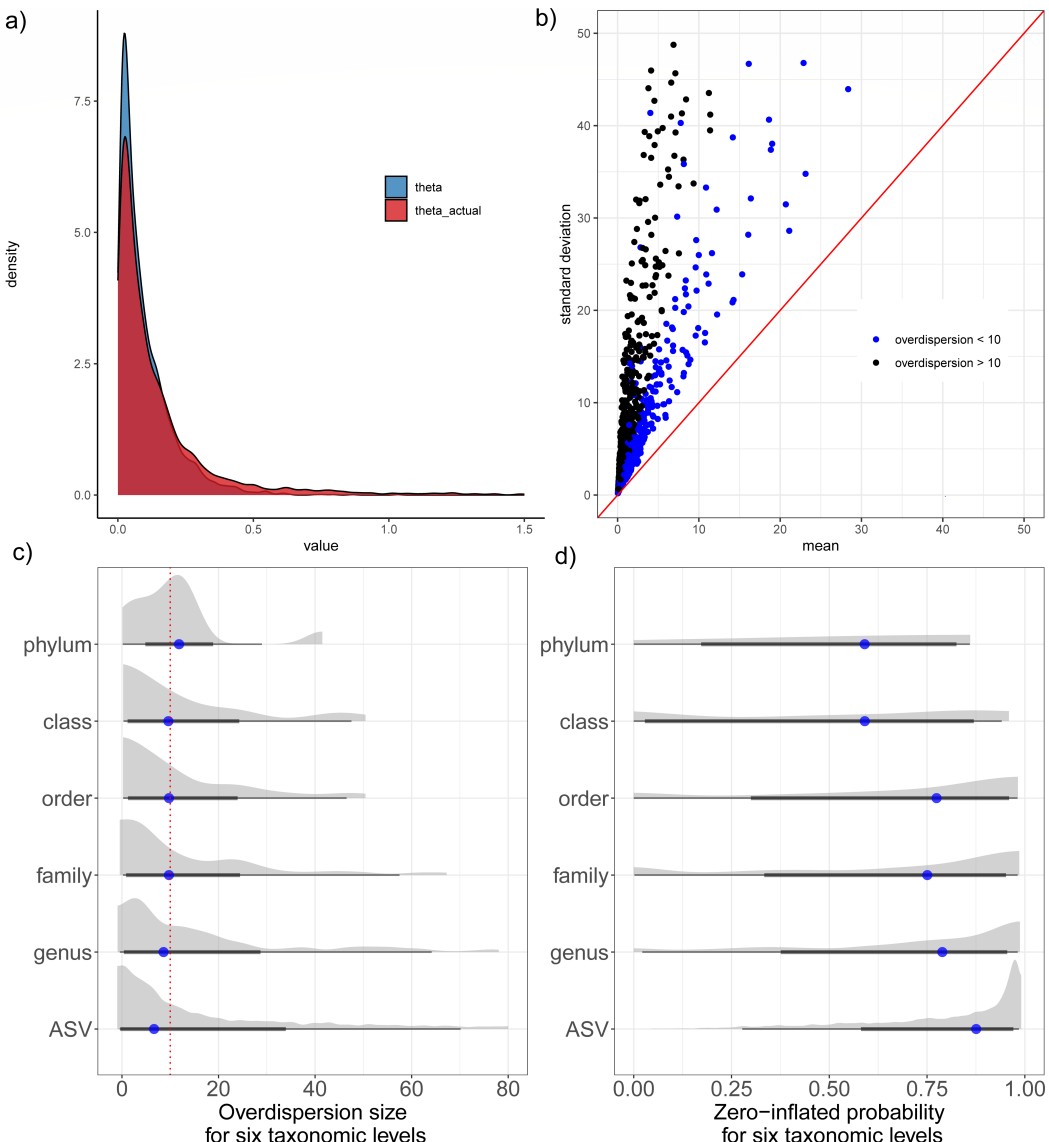

**FIG 3** Complexity of microbiome count data. (a) The discrete value of theta is estimated using the conversion equation. The blue part is the direct value, and the red part is the actual value; (b) Distribution of standard deviation and mean of microbial count data. Black dots indicate a value greater than 10, and blue indicates a value less than 10; (c, d) Overdispersion and zero-inflation parameters of the six taxonomic levels are counted separately. The dotted red intercept signifies a overdispersion value of 10.

relative abundance of these bacteria in obese or overweight people at the genus level (54–57). These results highlight the intricate nature of the gut microbial ecosystem, which means that the relative abundance of these so-called specific genera in obese individuals may vary geographically and ethnically.

Therefore, evaluating the ethnic and geographic influence on each ASV allowed us to uncover the diversity and distribution of these microbial species across different populations and regions. Then, we focus on ASVs with covariate effect sizes of any age or BMI exceeding 0.35. These ASVs have at least one pair of non-overlapping ethnogeographic interaction effects (see Fig. 5). We observed the effects of other ethnic groups by setting the Uyghur ethnic group, located at geographic position H, as the reference control group, in the context of the probability density of the posterior distribution for two covariates, ethnicity, and geographic location. We found the discrepancy of interaction between ethnicity and geography in different ASVs. *Ligilactobacillus* (ASV4)

was the most obvious, showing as follows: the positive effect of the Uyghur at location N (effect size: 0.50, 95% CI: [0.20, 0.80]), as well as negative effect of the Kazakh at location N, Uyghur with geographic S and W (effect size: −0.85, 95% CI: [−1.43, −0.26]; −1.53, 95% CI: [−2.58, −0.48]; −0.36, 95% CI: [−0.68, −0.04]). The *Bifidobacterium* (ASV1) showed similar results. Coincidentally, *Bifidobacterium* and *Ligilactobacillus* genera are the most common consumers of fermented foods and prebiotics (58). Thus, this prominent association is likely to be explained by regional diets or the heredity of local inhabitants (59). Indeed, we found that each cohort had their own distinctive eating habits, such as hand-fermented dairy products.

## DISCUSSION

We proposed and optimized the ZINB model, a method based on Bayesian parametric regression, specifically for modeling counting data in the presence of high zero inflation and high dispersion. We applied this model to studies aiming to dissect gut microbiome count data. The estimates based on the ZINB model provide a more nuanced understanding of how each microbial taxon is affected by covariates compared with conventional statistical tests alone. The ZINB ($\tau$) model has the advantage of being able to accommodate more complex data structures. Zero inflation is an inevitable property in gut microbiome count data, but some previous models, such as BhGLM and glmFit, ignore this property and primarily utilize a negative binomial distribution (60). Conversely, some other studies have used certain statistical models that may not be appropriate, producing results that overemphasize the zero-bloat nature of the gut microbiome data (12, 25, 43, 61). In the current study, our data processing results by using the ZINB model showed that only a small number of microbial taxa counts belonged to zero-inflated distributions, and the probability of zero inflation decreases continuously with an increase of taxonomic rank. So, it is important to carefully consider the properties of the data and choose appropriate models that can accurately capture the underlying structure of the data.

In addition to taking the properties of the zero-inflated characteristic into account, ZINB is also capable of considering the degree of dispersion. Compared with previous studies, we found that the degree of dispersion has a significant impact on the accuracy of model parameter estimation, and this effect can be mitigated by increasing the

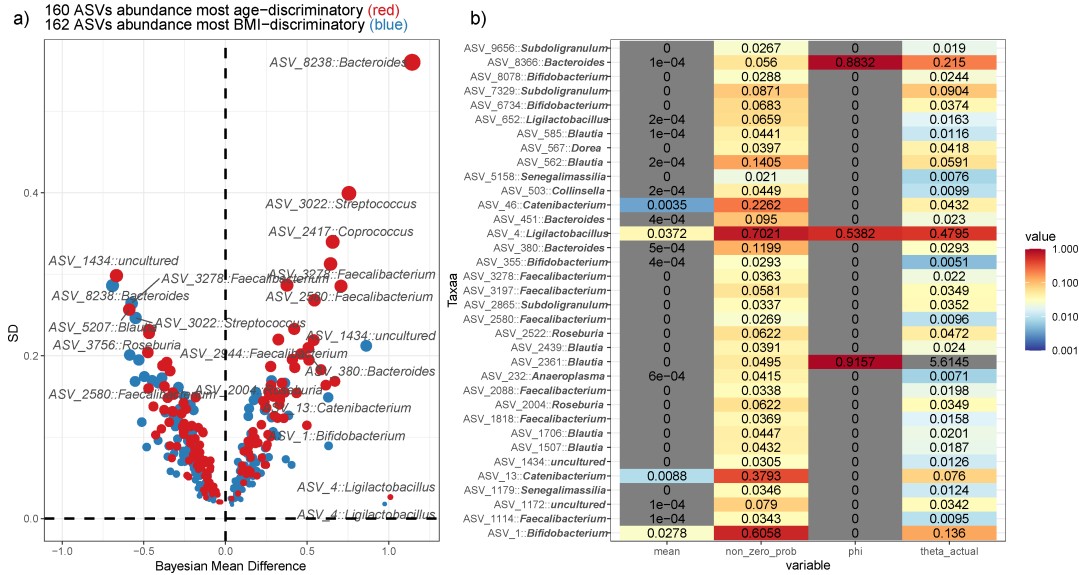

FIG 4 Bayesian effect size analysis was conducted to evaluate the impact of age and BMI on the abundance of taxa. (a) The graph plots the influence of age and BMI on taxa, with the effect size represented on the X-axis, and the error margin displayed on the Y-axis; (b) Attributes of taxonomic data (ASV), only when its covariate effect sizes of any age or BMI exceeding 0.35, are highlighted, including $\phi$, $\theta$, average relative abundance, and zero probability.

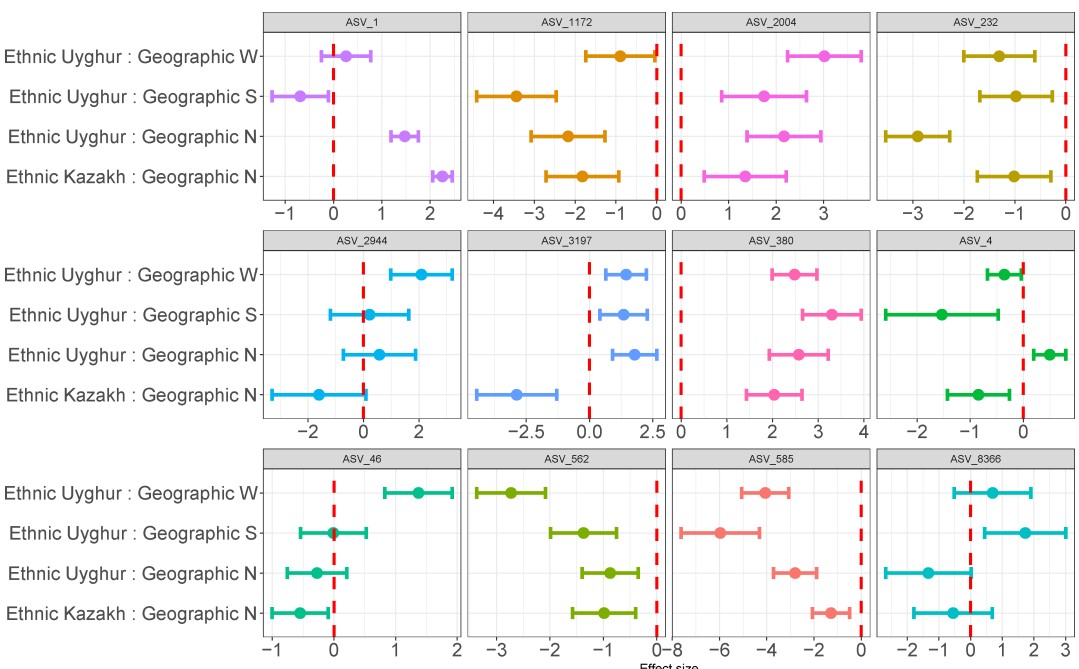

**FIG 5** Effect size of 12 taxa under the interaction of ethnicity and geography. Those covariate effect sizes of any age or BMI exceeding 0.35, and at least one pair of ethnogeographic interaction effects do not overlap. The red dotted line represents an effect size of zero.

sample size. Our study provides more detailed insights than previous studies into the relationship between dispersion and parameter estimation accuracy. Furthermore, we conducted a detailed discussion on each parameter of ZINB model. We provided insights into the interpretation and practical implications of each parameter in the model. This helps to provide a better understanding of how each parameter affects the model's performance and how to adjust them to achieve better results in different scenarios. In the initial step of our analysis, we eliminated taxa with fewer than 30 observed counts in the cohort that often lead to model sampling errors. Also, we observe that using the gamma distribution for the parameter $\tau$ was superior to the Gaussian distribution. Thus, our approach is adaptable since it permits the identification and estimation of the correlation between covariates and the abundance of each taxon. We developed a pipeline for analyzing microbial data using a zero-inflated count model was proposed, and applied it to a real cohort study. The pipeline emphasizes strategies for addressing the effect of covariate interactions, such as geography and ethnicity. Compared to adulthood, the microbial composition and body index in childhood tend to undergo relatively dramatic changes. To account for the rapid changes, we performed Z-score normalization on BMI values. Analyzing the variation in BMI values during early life stage requires a more nuanced approach than simply normalizing or centralizing (62) for covariate data.

As for our real-data analysis, we identified differentially abundant taxa in our model that were related to age or obesity, and certain taxa were identified as indicators of interaction with ethnicity or geography, such as the *Bifidobacterium* and *Ligilactobacillus*. Identifying interaction between differentially abundant taxa with ethnicity and geography would help to understand the factors that shape the characteristic human gut microbiome. It is already known that diet plays a decisive role in forming the gut microbiome of people with different genetic backgrounds. Particularly, the consumption of localized fermented foods inhabiting lactic acid bacteria may be related to certain bacterial taxa. Therefore, these findings have biological significance and help advance our understanding of the mechanisms underlying the gut microbiome. Moreover,

developing personalized interventions based on these findings could help promote a healthy gut microbiome in people with diverse backgrounds (63).

Although the zero-inflation negative binomial model is widely acknowledged as the most appropriate way to analyze count-based microbial community profiles, we observed some inconsistent behavior for estimating the probability of extra zeros (equation 3). This also underscores the potential reproducibility issues that can arise due to variations in algorithms, implementations, and computational environments, even when using the same underlying model (64, 65). It suggests that interpreting multiple implementations of the same statistical model for complex microbial community settings without an experimentally validated gold standard should be very cautious. Nevertheless, we also observe that the ZINB models suffer from a surplus of random number generation and calculations of myopia. The sensitivity of complex systems to small changes means that model design heavily relies on accurate initial conditions. As reported by Robert McCredie May, the founding father of discrete chaos theory (66), even the simplest logistic map exhibits an extraordinarily complicated dynamics. Overall, although there are several machine learning techniques recognized for their ability to analyze high-dimensional data and perform feature selection (67–69), the unique complexity of microbiome count data should not be overlooked when effectively analyzing studies. Therefore, as nonlinear trajectory-based methods from Bayesian become increasingly available, there is potential for future extensions that address sparse and irregular data, especially when there are multiple covariates (70). In addition, in microbiome studies, defining an appropriate magnitude for a reasonable and clinically meaningful effect size is unclear. Therefore, there is a need to develop frameworks that make the most of growing amount of microbiome–host interactomics data, facilitating the revelation of underlying biological mechanisms. In summary, the methods presented in our paper provide practitioners with a broad set of effective analytical strategies, with state-of-the-art reasoning capabilities, for identifying microbial associations from complex microbial community in the human microbiome studies.

## Conclusion

In this paper, we propose the ZINB model, which is a Bayesian generalized linear model capable of accounting for multivariate correlation structure, overdispersion, dimensionality issues, and zero inflation in the multi-ethnic child gut microbiome data. We discuss in detail the treatment of covariates, zero inflation and degree of dispersion in counting data. If these features of gut microbiome data are ignored, it can result in imprecise estimation of effect sizes and reduced statistical power. The results of this study can assist other research groups in making informed decisions when selecting a statistical analysis method.

## ACKNOWLEDGMENTS

We are grateful to Zhixuan Liang for her instruction in the grammar of the manuscript.

This work was funded by southern Xinjiang key industrial innovation and development project of Xinjiang Production and Construction Corps (2023AB050, 2024AB050), Joint Key Funds of the National Natural Science Foundation of China, and the Autonomous Region Government of Xinjiang, China (Grant No. U1903205).

J.H. conceived the idea for the manuscript. J.H. and Y.L. are responsible for code editing and figure creation in this study. F.T. and Y.N. are provided valuable guidance on the manuscript. All authors reviewed the manuscript and made corrections.

## AUTHOR AFFILIATIONS

[1]School of Food Science and Technology, Shihezi University, Shihezi, Xinjiang, China
[2]Key Laboratory of Xinjiang Special Probiotics and Dairy Technology, Shihezi University, Shihezi, Xinjiang, China

³State Key Laboratory of Food Science and Resources, Jiangnan University, Wuxi, Jiangsu, China
⁴School of Food Science and Technology, Jiangnan University, Wuxi, Jiangsu, China

## AUTHOR ORCIDs

Jian Huang  http://orcid.org/0009-0001-5085-8422
Yongqing Ni  http://orcid.org/0000-0003-4876-589X

## DATA AVAILABILITY

The simulation research and pipeline of the zero-inflated negative binomial (ZINB) model are now available for free on https://github.com/jianhuang525/ZINBmodel. For this paper's all ASV classification table, species covariate comments form, and sample information, see https://zenodo.org/records/11127872.

## ETHICS APPROVAL

This study involves human participants and the study was approved by the Ethics Committee of the First Affiliater Hospital, Shihezi University School of Medicine (2017-117-01). Participants gave informed consent to participate in the study before participation.

## ADDITIONAL FILES

The following material is available online.

### Supplemental Material

**Supplemental material (mSystems01345-24-s0001.docx).** Appendix S1 and Fig. S1 to S5.
**Supplemental tables (msystems01345-24-s0002.xlsx).** Tables S1 to S4.

### Open Peer Review

**PEER REVIEW HISTORY (review-history.pdf).** An accounting of the reviewer comments and feedback.

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
