## [Reviewer comments · mSystems]

Association of Body Index with Fecal Microbiome in Children Cohorts with Ethnic-Geographic Factor Interaction: Accurately Using a Bayesian Zero-inflated Negative Binomial Regression Model

jian huang, Yanzhuan Lu, Fengwei Tian, and Yongqing Ni

Corresponding Author(s): Yongqing Ni, Shihezi University

Review Timeline:

Submission Date:

October 9, 2024

Accepted:

October 24, 2024

Editor: Cheng Gao

Reviewer(s): Disclosure of reviewer identity is with reference to reviewer comments included in decision letter(s). The following individuals involved in review of your submission have agreed to reveal their identity: Dong Chen (Reviewer #1); Luciana Damascena Silva (Reviewer #2)

Transaction Report:

DOI: <https://doi.org/10.1128/msystems.01345-24>

Re: mSystems01345-24 (Association of Body Index with Fecal Microbiome in Children Cohorts with Ethnic-Geographic Factor Interaction: Accurately Using a Bayesian Zero-inflated Negative Binomial Regression Model)

Dear Dr. Yongqing Ni:

Your manuscript has been accepted, and I am forwarding it to the ASM production staff for publication. Your paper will first be checked to make sure all elements meet the technical requirements. ASM staff will contact you if anything needs to be revised before copyediting and production can begin. Otherwise, you will be notified when your proofs are ready to be viewed.

Sincerely,
Cheng Gao
Editor
mSystems

Reviewer #1 (Comments for the Author):

I have carefully reviewed the revised manuscript and found that the authors have addressed my previous concerns effectively. The changes made have significantly improved the quality and clarity of the paper. The research presented is sound, and the data are well-analyzed and presented. The manuscript is now well-organized, and the arguments are clearly articulated.

The authors have responded to each of my points with appropriate revisions and have provided a thorough explanation for their changes. The additional data included in the revised version strengthen the conclusions drawn from the study. The references have been updated, and the formatting is in line with the journal's guidelines.